# Quantitative Health Risk Assessment of the Chronic Inhalation of Chemical Compounds in Healthcare and Elderly Care Facilities

**DOI:** 10.3390/toxics10030141

**Published:** 2022-03-15

**Authors:** Anaïs Colas, Alexandre Baudet, Pierre Le Cann, Olivier Blanchard, Jean-Pierre Gangneux, Estelle Baurès, Arnaud Florentin

**Affiliations:** 1Faculté de Médecine, Université de Lorraine, F-54505 Vandoeuvre-les-Nancy, France; arnaud.florentin@univ-lorraine.fr; 2CHRU-Nancy, F-54505 Vandoeuvre-les-Nancy, France; alexandre.baudet@univ-lorraine.fr; 3Faculté D’odontologie, Université de Lorraine, F-54505 Vandoeuvre-les-Nancy, France; 4APEMAC, Université de Lorraine, F-54505 Vandoeuvre-les-Nancy, France; 5EHESP School of Public Health, Inserm, IRSET (Institut de Recherche en Santé, Environnement et Travail)—UMR_S 1085, Université de Rennes, F-35000 Rennes, France; pierre.lecann@ehesp.fr (P.L.C.); olivier.blanchard@ehesp.fr (O.B.); jean-pierre.gangneux@univ-rennes1.fr (J.-P.G.); estelle.baures@ehesp.fr (E.B.); 6Laboratoire de Parasitologie-Mycologie, CHU-Rennes, F-35000 Rennes, France

**Keywords:** health risk assessment, indoor air pollution, chronic inhalation, chemical compounds, healthcare facility, hospital, nursing home

## Abstract

Previous studies have described the chemical pollution in indoor air of healthcare and care facilities. From these studies, the main objective of this work was to conduct a quantitative health risk assessment of the chronic inhalation of chemical compounds by workers in healthcare and elderly care facilities (hospitals, dental and general practitioner offices, pharmacies and nursing homes). The molecules of interest were 36 volatile and 13 semi-volatile organic compounds. Several professional exposure scenarios were developed in these facilities. The likelihood and severity of side effects that could occur were assessed by calculating the hazard quotient for deterministic effects, and the excess lifetime cancer risk for stochastic effects. No hazard quotient was greater than 1. Three compounds had a hazard quotient above 0.1: 2-ethyl-1-hexanol in dental and general practitioner offices, ethylbenzene and acetone in dental offices. Only formaldehyde presented an excess lifetime cancer risk greater than 1 × 10^−5^ in dental and general practitioner offices (maximum value of 3.8 × 10^−5^ for general practitioners). The health risk for chronic inhalation of most compounds investigated did not appear to be of concern. Some values tend to approach the acceptability thresholds justifying a reflection on the implementation of corrective actions such as the installation of ventilation systems.

## 1. Introduction

Indoor air quality is a major issue. In healthcare facilities, especially in hospitals, the indoor air is studied in the field of infection control and prevention. Apart from the risk of infection, air pollution is little studied in the healthcare and care facilities. The most studied indoor environments are public establishments such as schools, office buildings, and dwellings. Healthcare and care facilities receive every day a large number of patients and workers but only a few studies have been carried [1].

Indoor air contains a mixture of chemical and microbiological compounds that can affect the health of exposed people [2,3]. People are exposed to volatile organic compounds (VOCs) and semi-volatile organic compounds (SVOCs) through inhalation, skin contact, and ingestion depending on their gaseous or particulate form [4]. It is now well established that organic compounds may lead to various health troubles [5,6].

Knowledge of indoor air quality in facilities for both public and worker uses is important, especially where the public is potentially vulnerable such as in healthcare and elderly care facilities. Moreover, senior citizens spent more than 80% of their time indoors at home [7]. The workers are also widely exposed to indoor air in work environments as they spend more than 30% of their time working indoors [8].

In healthcare facilities, the use of chemical products such as disinfectants and detergents may contribute to higher concentrations of some compounds in the air compared with other establishments [9]. Chemicals may also be emitted from building materials and from the outdoor environment [10]. To limit the accumulation of chemical and microbiological compounds, hospitals have efficient ventilation systems designed to improve air quality. However, a large part of other private healthcare facilities, such as dental and general practitioner offices, do not have such ventilation systems.

Very poor studies have quantified the health risk assessment of the chronic inhalation of chemical compounds in healthcare and elderly care facilities. Almeida-Silva et al. only assessed the elderly’ daily exposure to air pollutants in elderly care facilities [11]. Hong et al. conducted a health risk assessment to determine the chronic health effects of VOCs to dental professionals but only in a dental clinic [12]. Hwang et al. conducted a health risk assessment in hospitals, geriatric hospitals, elderly care facilities and postnatal care centers for non-carcinogenic risk due to exposure to formaldehyde. Although classified as carcinogenic, formaldehyde health risk assessment for carcinogenic risk has not been studied [13,14]. In a first study [15], we described the chemical pollutants in the indoor air of two French hospitals. The indoor air pollution of the French hospitals was low, probably due to the central air conditioning systems. In a second study [16], we described the chemical pollutants in the indoor air of private healthcare and elderly care facilities (dental and general practitioner offices, pharmacies and nursing homes) in two French urban areas. Indoor air of these facilities contained a complex mixture of chemical, particulate and microbiological compounds. The most frequently quantified compounds were alcohols (ethanol and isopropanol) originating mainly from healthcare activities. Indoor air of private healthcare and elderly care facilities showed higher pollution compared to hospital settings.

From these two previous studies, the main objective was to conduct a quantitative health risk assessment of the chronic inhalation of chemical compounds in the healthcare and elderly care facilities studied (hospitals, dental and general practitioner offices, pharmacies and nursing homes).

## 2. Materials and Methods

### 2.1. Hazard Identification and Compounds of Interest

The compounds of interest were the organic compounds studied in the two previous studies [15,16]: 36 VOCs and 13 SVOCs (Table 1). The choice of these molecules was based on a previous study which identified compounds that presented a danger and could generate adverse health effects. This methodology, developed by Berrubé et al., was based on a combination of classes of exposures and classes of hazard that produces a risk matrix. Criteria used to determine the exposure classes were quantity and frequency of the chemical’s use, volatility and the type of collective protective equipment associated in each area. Criteria used to determine the hazard classes were classification as carcinogenic, mutagenic, or toxic to reproduction; the existence of a toxicological reference values (TRV); occupational exposure limit values; and risk phrases (official risk descriptions of the products) [17].

### 2.2. Dose-Response Assessment

Dose-response assessment characterizes the quantitative relationship between exposure and the occurrence of adverse health effects (usually determined in toxicity studies). The route of exposure of interest was chronic inhalation. Co-exposures were not considered in this study.

In this study, toxicological reference values (TRV) used were those established by health agencies (The French Agency for Food, Environmental and Occupational Health and Safety (ANSES, Maisons-Alfort, France), the United States Environmental Protection Agency (US EPA, Washington, DC, USA), the Agency for Toxic Substances and Disease Registry (ATSDR, Atlanta, GA, USA), etc.). The choice of VTR was made according to the recommendations of the French Ministry of Ecology, Sustainable Development and Energy to conduct health risk assessment [18]. Selected TRVs and compounds of interest’s toxicity are presented in Appendix A [19].

### 2.3. Exposure Assessment

The health risk assessment method used here is a quantitative approach based on indoor air samples performed in several rooms of healthcare and care facilities (Table 2). Two French cities were investigated, each measurement was conduct over a week in summer and then repeated in winter. Sampling and analyses were described in a previous paper [15]. In brief, passive sampling was used to collect the aldehydes using 2,4-DNPH cartridges (Radiello™) (SUPELCO^®^ by Sigma-Aldrich, St. Louis, MO, USA), and active sampling was used to collect the other VOCs using Carbopack™/Carboxen^®^ tube (SUPELCO^®^ by Sig-ma-Aldrich, St. Louis, MO, USA) and to collect the SVOCs using polyurethane foam (PUF) and quartz filter (University Research Glassware, Chapel Hill, NC, USA). Aldehydes, other VOCs and SVOCs were simultaneously analyzed in air samples by chemical desorption and high-performance liquid chromatography with diode array detection (HPLC/DAD), thermal desorption (TD) and gas chromatography/mass spectrometry (GC/MS), pressurized liquid extraction (PLE) and gas chromatography/tandem mass spectrometry (GC/MS/MS), respectively.

Several exposure scenarios for different professions in healthcare and elderly care facilities were developed from observational data and questionnaires on the habits of the frequentation of premises, including an endoscope Disinfection Unit’s technician, a laboratory technician, a care unit’s nurse, an anesthesia Care Unit’s nurse, a dental surgeon and a dental assistant, a general practitioner a nursing home’s nurse, nursing assistant, physiotherapist and resident, a pharmacist, and a pharmacy technician.

In hospitals, the concentrations of organic compounds sampled in the reception hall were considered as the basic exposure value and were used to calculate the exposure in places where ambient concentrations were not measured (rest room, non-care unit activity, …). In private healthcare and elderly care facilities, the waiting rooms or common room concentrations were considered as the basic exposure value.

### 2.4. Risk Characterization

The likelihood and severity of side effects that could occur were assessed by calculating the hazard quotient (HQ) for deterministic effects (threshold chronic effects) and the excess lifetime cancer risk (ELCR) for stochastic effects (non-threshold effects).

The HQ represents the ratio of the exposure concentration of a compound to the concentration at which no adverse health effects are expected following chronic inhalation (TRV). The exposure concentration used was the mean concentration measured in each type of room for each compound multiplied by the mean inhaled volume of an individual and weighted by the time spent in rooms investigated (40-year career) over a 70-year lifetime. If an organic compound concentration measured was below the quantification threshold, this threshold was used to calculate the HQ (worse-case scenario). A HQ > 1, indicates a possibility that some non-carcinogenic effects may occur.

The average inhaled volume varies according to physical activity. The values used were those given by the US EPA [20]. The activity was considered low for general practitioners, dentists, dental assistants, pharmacists, pharmacy technician, nursing home residents, care unit’s nurse and anesthesia care unit’s nurse: 0.6 m^3^/h; mixed (low: 0.6 m^3^/h and moderate: 2.1 m^3^/h) for nursing home’s nurses, physiotherapists and nurse assistants according to the following conditions. Nurse assistants: moderate activity for half the time spent in residents’ room, low activity for the rest of the time; nurses and physiotherapists: moderate activity for 1/3 of the time spent outside the resident’s room, low activity for the rest of the time.

The ELCR estimate the lifetime excess cancer risk which is the exposure concentration multiplied by the cancer potency factor (TRV for cancer risk).

## 3. Results

### 3.1. Organic Compounds Concentrations

The most-quantified VOCs with the highest mean concentration measured in all facilities were ethanol (358.7 µg/m^3^), isopropanol (27.4 µg/m^3^) and acetone (26.3 µg/m^3^). The most-quantified aldehydes were formaldehyde (11.9 μg/m^3^) and acetaldehyde (6.4 μg/m^3^). Aliphatic and halogenated hydrocarbons were found in low concentrations. Regarding SVOCs, no pyrethroids were detected. Among the phthalates, DEP, DBP and DiBP were identified in all facilities. The detailed results are described in other papers [15,16].

The concentrations of the organic compounds measured in dental and general practitioner offices and in pharmacies were generally higher than those measured in the hospitals with the exception of ether and toluene where significant concentrations were measured in the parasitology and mycology laboratories (toluene: 11.0 μg/m^3^; ether: 40.7 μg/m^3^).

### 3.2. Exposure Assessment

Thirteen exposure scenarios were developed from questionnaires or observational data. Due to the lack of response from the general practitioner, data from a national survey on the working time of general practitioners in France were used. All scenarios are described in Table 3.

### 3.3. Deterministic Effects

In private healthcare and elderly care facilities, no HQ was greater than 1 and only three molecules had a HQ above 0.1: ethylbenzene, 2-ethyl-1-hexanol and acetone. The highest HQ had a value of 0.24 and concerned dental surgeons’ exposure to ethylbenzene by chronic inhalation. Regarding SVOCs, all HQs were below 1 × 10^−3^. The highest HQs of VOCs and SVOCs are presented by occupation in Figure 1. All results of HQs are presented in Appendix A. Dental offices’ HQs were higher than other facilities.

Limonene and isovaleraldehyde are among the molecules for which the HQs could not be calculated due to the lack of an available TRV.

In hospitals, no HQ was greater than 1 and only five molecules had a HQ above 0.01: ethylbenzene, 2-ethyl-1-hexanol, acetone, acetaldehyde and propionaldehyde. The higher HQ had a value of 0.06 and concerned laboratory technicians’ exposure to ethylbenzene by chronic inhalation. Regarding SVOCs, all HQs were below 1 × 10^−3^. The laboratory technician’s HQ were higher than the other healthcare workers.

### 3.4. Stochastic Effects

Many ELCR could not be calculated due to the lack of available TRV for non-threshold effects.

In private healthcare and elderly care facilities, all of the formaldehyde ELCRs (except for the nursing home’s resident) were greater than 1 × 10^−5^. Only two other molecules had an ELCR above 1 × 10^−6^: benzene and acetaldehyde. The highest ELCR had a value of 3.8 × 10^−5^ and concerned general practitioner’s exposure to ethylbenzene by chronic inhalation. Concerning the SVOCs, only the ELCRs of DEHP could be calculated and were less than 1.5 × 10^−8^. The highest ELCRs are presented by occupation in Figure 2. All results of ELCR are presented in Appendix A. The nursing home’s resident had slightly lower values than the healthcare workers.

## 4. Discussion

### 4.1. Methodology

Two previous studies have measured ambient concentrations of various pollutants in the indoor air of healthcare and elderly care facilities [15,16]. Most concentrations in private healthcare and elderly care facilities were higher than hospitals. Beyond potential differences in the practice and use of chemicals, ventilation systems in hospitals allows the renewal of the air more efficiently and most likely contribute to the reduction of the organic compounds concentrations.

The exposure assessment in this study was based on an indirect method. The exposure concentrations were calculated from ambient concentrations measured in certain rooms. In the context of an individual approach, it would have been more appropriate to measure the absorbed dose directly using portable sensors.

Only a few rooms were investigated per type of facility. After exposure data collection, we noticed that some professionals spent time in other non-investigated rooms, for example nursing care rooms in nursing homes. To improve this study, another sampling campaign in private healthcare and elderly care facilities was planned, but the health restrictions had so far not made it possible to continue the investigations. Ambient concentrations of the common rooms were then used for the time spent in the rooms not investigated. This method introduces a bias in the exposure assessment but it also limits the undervaluation of exposures. Taking costs and technical feasibility into account, the indirect approach for the main parts remained an appropriate alternative within the framework of a preliminary study.

Some exposure scenarios are based on declarative data from a single subject and are therefore not representative. The resident scenario considers the average time spent in the room is 23 h. It is obvious that other residents spend several hours a day in the common room, but due to a lack of data, this type of scenario could not be studied. In nursing homes, we noted that for almost all of the VOCs and SVOCs, the concentrations were higher in the rooms. Our resident scenario can then be considered as the worse-case scenario.

The exposure concentrations are also based on the volume of air inhaled. The volume value depends on the level of physical activity and can vary by a factor of six between low activity and intense activity [20]. It was decided to consider the low level for the majority of professionals and the resident, but adjustments were made for nursing homes’ staff. The lack of personnel in these facilities leads to pooling professionals on different floors. To consider the many comings and goings within the buildings, it was decided to consider physical activity as moderate for a portion of the time spent outside residents’ rooms for each professional. These choices were made arbitrarily based on our field experience in nursing homes as it seemed appropriate to modulate the level of activity according to the tasks performed to best approach reality.

The main limitation remains the low number of structures investigated, thus leading to a lack of representativeness: both in the average calculation of ambient concentrations but also in the construction of space-time budgets. These results can only be interpreted on a local scale and should not be extrapolated as they stand to all healthcare facilities. This data should be consolidated by replicating the sampling campaigns in other structures and increasing the number of healthcare workers questioned/observed. It is possible that certain indoor pollutants of interest in healthcare facilities were not sought during ambient measurements. An additional screening should have taken place to look for other compounds (disinfectants or even solvents and resins used during dental care for example). Comparison with outdoor measurements should also be considered since it would make it possible to assess pollution levels and thus estimate the share of transfer of outdoor pollution, often the cause of high ambient concentrations [22].

### 4.2. Deterministic Effects

Due to the lack of TRV, some HQs could not be calculated. Risk assessment could not be conducted for limonene and isovaleraldehyde, which represent some of the highest exposure scenarios for the pharmacist and the pharmacy technician, the endoscope disinfection unit’s technician, and the care unit’s nurse. Mean limonene concentrations measured in dental offices and pharmacies were higher than those measured in building offices [23] but less than or equal to those in dwellings [24]. Limonene effects, present in some cleaning products and cosmetics, are not well documented. This compound is considered an irritant but does not appear to cause systemic effects at ambient concentrations measured in indoor air [25].

Among the HQs calculated, none was greater than 1: the health risk for chronic inhalation of the VOCs and SVOCs investigated does not seem to be of concern. Nevertheless, some values tend to approach 1 and these values only concerned dental and general practitioner offices. Three VOCs had HQs greater than 0.1 for dental office professionals: ethylbenzene, 2-ethyl-1-hexanol and acetone.

The ethylbenzene HQs values are explained by a very high ambient concentration measured in summer in the consultation room of a dental office (more than 26.1 μg/m^3^) while much lower quantities were measured in winter and in the other dental office (less than 1.3 μg/m^3^). This geographic and seasonal variability could be explained by transient outdoor pollution linked to road traffic, a source of aromatic hydrocarbon emissions [10]. The potential occurrence of benzene’s ototoxic effects would not be directly related to the activity but to the geographical location and the proximity of a source of external pollution.

The acetone HQs values are explained by greater exposure of professionals in dental offices (higher ambient concentrations of acetone than in other facilities). However, after a bibliographic search, no source of exposure was identified among the various products used in everyday practice in dental offices. To be exhaustive, it would have been necessary to carry out an inventory of the products used specifically in these dental offices and study their compositions. Although not demonstrated here, the occurrence of neurological effects related to chronic inhalation of acetone seems more likely for dental surgeons and dental assistants than for the other healthcare workers. In a dental clinic, Hong et al. identified a healthcare risk for dental professionals regarding acetone with a HQ = 4.1 [12].

The 2-ethyl-1-hexanol HQs were greater than 0.1 for healthcare workers in all private healthcare and elderly care facilities. This compound is usually used as a raw material for the manufacture of DEHP. In the indoor environment, flooring can be a significant source of emissions. It can also be used as a fragrance in some cleaning products and detergents [26]. It is important to specify that only 20% of the ambient concentrations measured were above the detection threshold in private healthcare and elderly care facilities. These results therefore illustrate the worst-case scenario and are overestimated; the ambient concentrations are probably below the threshold value used for the calculations. Moreover, the concentrations measured were not higher than those measured in other public places [27]. The value used as a TRV was a provisional value proposed by the US EPA for sub-chronic exposure [28]. Given the limits described above, it is difficult to conclude on the potential health risk of chronic exposure by inhalation to 2-ethyl-1-hexanol for healthcare workers in private healthcare and elderly care facilities.

In a dental clinic, Hong et al. quantified a few VOCs and found HQs less than 1 for methylate, chloride, styrene, toluene and acetone, but HQs upper than 1 for methyl methacrylate and acetone (16.4 and 4.1, respectively) [12]. In comparison with French dwellings, most of the VOCs and SVOCs were measured in higher concentrations at home [24,29]. Regarding SVOCs, the HQs were less than 1 [30]. Regarding VOCs, their high concentrations in French dwellings were associated with an increasing prevalence of asthma and rhinitis in adults [31].

### 4.3. Stochastic Effects

A carcinogenic effect is suspected or proven in humans for halogenated hydrocarbons [32]. Inhalation is the main exposure route for these chemicals, but in our assessment the concentrations measured were very low. None of the ELCRs exceeded the acceptability threshold of 1 × 10^−5^.

Regarding aromatic hydrocarbons, naphthalene and ethylbenzene, classified as possible carcinogens [33,34], had low ELCRs. Benzene, on the other hand, showed higher ELCRs among healthcare workers but still below the acceptability threshold. Classified as a proven carcinogen after evidence of an association between exposure to benzene and the occurrence of leukemia [35], its use has since been strictly regulated in the professional environment. Indoor air pollution is mainly linked to combustion (smoking, heating, etc.) and the transfer of outdoor air pollution [36]. Although some materials and products still contain benzene, it has largely been substituted by other components. In healthcare facilities, only certain specific laboratory products could constitute a significant source of exposure. Here, we have not demonstrated a higher concentration in laboratories.

Aldehydes, when inhaled, are suspected of being involved in nasopharyngeal cancer. Commonly used in products for its biocidal and disinfectant properties, formaldehyde (proven carcinogenic potential) is now subject to use restrictions with a specific marketing authorization procedure [14]. However, indoor air pollution by the release of materials or the use of household products still persists. This phenomenon was also observed in this study since the ELCRs of formaldehyde were greater than 1 × 10^−5^ for healthcare workers in private healthcare and elderly care facilities. The absence of risk linked to the inhalation of formaldehyde on the various sites cannot be demonstrated. As described previously, the ambient concentrations measured were lower than those in dwellings [24,29]. Exposure in the workplace therefore does not seem to be greater than at home. In 2008, the French Agency for Environmental and Occupational Health Safety concluded that housing mainly contributed to exposure to formaldehyde in indoor environments but considered the risk of carcinogenesis linked to the presence of formaldehyde in these environments as negligible for the general population [37]. In France, the guideline value for indoor air is set at 30 μg/m^3^ for long-term exposure defined as being greater than one year. The average concentration measured in general practitioner offices (31.45 μg/m^3^) exceeds the guideline value. According to the recommendations of the French High Council of Public Health, putting in place an action plan to improve air quality in these offices is needed [38]. This plan is all of the more needed as the regulations provides for a drop in the guideline value to 10 μg/m^3^ by 2023. All of the other private healthcare and elderly care facilities of this study would thus be concerned. None of the general practitioner offices investigated had a controlled mechanical ventilation. A study on the air quality of establishments open to the public showed that natural ventilation by opening windows was not sufficient to reduce the concentrations of certain pollutants, in particular formaldehyde [39]. The results of this evaluation are therefore in favor of the installation of a ventilation system in general practitioner offices. In the hospital study [15], the samples were collected using a different methodology and cannot be interpreted in the same way. Yet, the formaldehyde concentrations were significantly lower (median concentrations comprised between 1.1 and 9.1 μg/m^3^).

In a dental clinic, Hong et al. found ELRC of 1.1 × 10^−2^ for formaldehyde, 5.6 × 10^−4^ for chloroform, 2.4 × 10^−6^ for methyl chloride, and 0.1 × 10^−6^ for benzene [12]. In comparison with others European indoor environments, the ELCRs calculated in this study were lower [40]. In comparison with French dwellings, no ELCRs were greater than 1 × 10^−5^ for SVOCs [30]. In China dwellings, the highest ELCRs were found for formaldehyde (6.6 to 7.2 × 10^−5^) [41].

Due to their significant use, alcohols must be subject to special monitoring. The absence of TRVs for the inhalation of these alcohols, especially ethanol, did not allow a health risk assessment. The ambient measurements were carried out before the health crisis and therefore before the increased use of hydroalcoholic solutions. Ambient concentrations, already high since ethanol was the compound with the highest levels, could be increased.

It is important to mention the impact of the COVID-19 pandemic on indoor air quality. In addition to the widespread use of hydroalcoholic solutions for hand hygiene, the time spent indoors has significantly increased (lockdowns, generalization of teleworking, etc.). During lockdowns, chemical quality of the air in dwellings has deteriorated due to time spent indoors [42] while the quantity of PM_2.5_ decreased due to outdoor air polluting activities cessation [43,44]. The spread of the virus has caused many changes in the patient care organization (waiting rooms removal, introduction of systematic disinfection between two patients and rooms’ ventilation frequency [45,46]). It is very likely that this new organization will last over time and will have an impact on the air quality of healthcare facilities. Regular renewal of indoor air would limit the accumulation of some chemicals such as VOCs, which are present in lesser quantities in outdoor air [27,47]. For other chemicals, an external transfer could take place (proximity to polluted sites or activities generating harmful compounds) [22].

The management of indoor air quality has experienced an upheaval during the pandemic. In France, new recommendations were issued in 2021 for all establishments open to the public [48]. The French High Council of Public Health recommends the development of an environmental strategy for controlling air quality through aeration and ventilation of premises. This plan provides for regular ventilation of occupied enclosed spaces and CO_2_ concentrations monitoring as a tracer of air renewal and of indoor air quality.

## 5. Conclusions

This study is the first to assess the health risk of chemical contamination of different VOCs and SVOCs in hospitals, dental and general practitioner offices, pharmacies, and nursing homes. Indoor air contained a complex mixture of many pollutants but the organic compounds concentrations sampled in indoor air revealed low chemical pollution of the facilities investigated. This mixture of pollutants was mainly composed of alcohols (ethanol and isopropanol), acetone, aldehydes (mainly formaldehydes and acetaldehyde), toluene and limonene. Concentrations in private healthcare and elderly care facilities were generally higher than in hospitals, similar to those found in building offices but lower than those measured in dwellings.

The calculation of HQs did not highlight a health risk for the deterministic effects. Concerning the stochastic effects, only formaldehyde presented ELCRs above the acceptability threshold of 1 × 10^−5^ for all healthcare workers in private healthcare and elderly care facilities, probably due to a lack of ventilation.

## Figures and Tables

**Figure 1 toxics-10-00141-f001:**
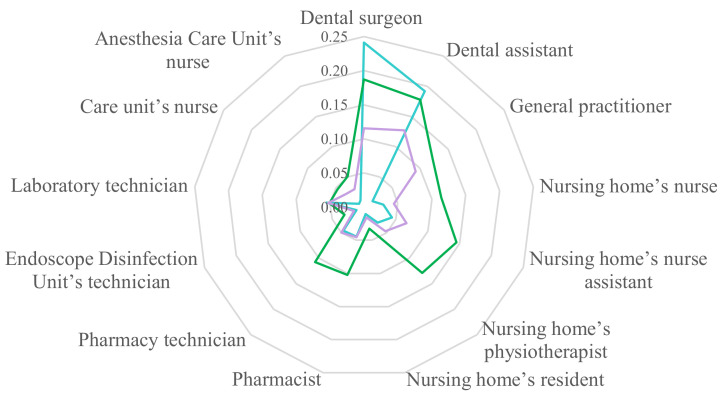
Hazard quotients of ethylbenzene (blue), 2-ethyl-1-hexanol (green) and acetone (purple).

**Figure 2 toxics-10-00141-f002:**
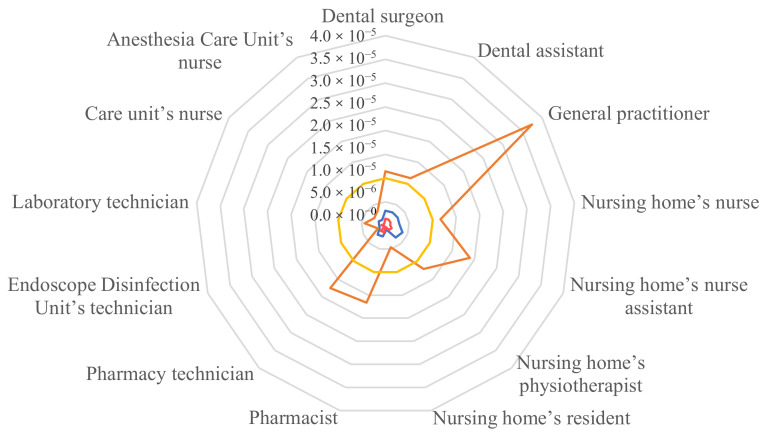
Excess lifetime cancer risks of benzene (blue), formaldehyde (orange) and acetaldehyde (red) against the acceptability threshold (yellow).

**Table 1 toxics-10-00141-t001:** Compounds of interest.

	Organic Compound
Volatile Organic Compounds (VOCs) (36)	
Aromatic hydrocarbons (9)	benzene, ethylbenzene, styrene, toluene, o-xylene, mp-xylenes, 1,2,4-trimethylbenzene, naphthalene, phenol
Aliphatic hydrocarbons (3)	n-decane, n-undecane, n-heptane
Halogenated hydrocarbons (7)	1,1,1-trichloroethane, 1,4-dichlorobenzene, trichloroethylene, tetrachloroethylene, bromodichloromethane, tribromomethane, trichloromethane
Alcohols (4)	2-ethyl-1-hexanol, ethanol, isopropanol, n-propanol
Ketones (2)	acetone, 2-butanone
Terpenes (1)	limonene
Ethers (3)	ether, 2-ethoxyethanol, 2-butoxyethanol
Aldehydes (7)	formaldehyde, acetaldehyde, propionaldehyde, butyraldehyde, isovaleraldehyde, valeraldehyde, hexaldehyde
**Semi-volatile organic compounds (SVOCs) (13)**	
Phthalates (6)	di(2-ethylhexyl)phthalate (DEHP), diethylphthalate (DEP), dibutylphthalate (DBP), diisobutyltphthalate (DiBP), benzylbutylphthalate (BBP), diisononylphthalate (DiNP)
Musk (2)	tonalide, galaxolide
Pyrethroids (5)	cyfluthrine, cypermethrine, deltamethrine, permethrine, tetramethrine

**Table 2 toxics-10-00141-t002:** Sampling sites and periods.

Facilities (Number)	Rooms	Sampling Period
Hospitals (2)	Patient room Reception hall Parasitology and mycology laboratory Plaster cast room Post-anesthesia care unit Nursing care room Flexible endoscope disinfection unit	Summer 2014 Winter 2015
Dental offices (2)	Sterilization room Waiting room Treatment room	Summer 2018 Winter 2019
General practitioner offices (2)	Waiting room Consulting room	
Pharmacies (2)	Commercial space Storage room	
Nursing homes (4)	Resident’s bedroom Common room (refectory or lounge)	

**Table 3 toxics-10-00141-t003:** Exposure scenarios.

Exposure Scenarios	Number of Respondents	Number of Days Worked per Week	Number of Weeks Worked per Year	Mean Daily Time Spent in the Different Premises (Hours)
Dental surgeon	3 (questionnaire)	4.2	47.0	Treatment room: 7.0 Sterilization room: 0.2 Waiting room: 0.0 Other rooms: 0.7
Dental assistant	3 (questionnaire)	4.0	45.0	Treatment room: 5.5 Sterilization room: 0.7 Waiting room: 0.2 Other rooms: 2.5
General practitioner	2161 (national survey data [21])	5.0	46.7	Consulting room: 10.0 Other rooms: 1.4
Nursing home’s nurse	2 (questionnaire)	3.0	47.0	Common room: 1.7 Resident’s room: 6.0 Other rooms: 4.3
Nursing home’s nurse assistant	2 (questionnaire)	3.0	47.0	Common room: 3.0 Resident’s room: 7.0 Other rooms: 1.0
Nursing home’s physiotherapist	1 (questionnaire)	5.0	44.0	Common room: 0.0 Resident’s room: 3.0 Other rooms: 4.5
Nursing home’s resident	1 (questionnaire)	7.0	52.0	Common room: 0.5 Resident’s room: 23.0 Other rooms: 0.5
Pharmacist	1 (questionnaire)	5.0	47.0	Commercial space: 5.0 Storage room: 2.0
Pharmacy technician	1 (questionnaire)	5.0	47.0	Commercial space: 6.5 Storage room: 1.0
Endoscope Disinfection Unit’s technician	5 (observational data)	5.0	44.0	Flexible Endoscope Disinfection Unit: 5.0 Other rooms: 3.3
Laboratory technician	5 (observational data)	5.0	44.0	Laboratory: 3.8 Other rooms: 4.5
Care unit’s nurse	5 (observational data)	5.0	44.0	Nursing Care Room: 3.0 Patient Room: 4.0 Other rooms: 1.3
Anesthesia Care Unit’s nurse	5 (observational data)	5.0	44.0	Post-Anesthesia Care Unit: 6.0 Other rooms: 2.3

## Data Availability

The data presented in this study are available in Baurès, E.; Blanchard, O.; Mercier, F.; Surget, E.; le Cann, P.; Rivier, A.; et al. Indoor air quality in two French hospitals: Measurement of chemical and microbiological contaminants. *Sci. Total Environ.*
**2018**, *642*, 168–79, and in Baudet, A.; Baurès, E.; Guegan, H.; Blanchard, O.; Guillaso, M.; Le Cann, P.; et al. Indoor air quality in healthcare and care facilities: chemical pollutants and microbiological contaminants. *Atmosphere*
**2021**, *12*, 1337.

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
