# Peer review of "Quantitative Health Risk Assessment of the Chronic Inhalation of Chemical Compounds in Healthcare and Elderly Care Facilities"

_toxics, 2022, doi:10.3390/toxics10030141_

Round 1

Reviewer 1 Report

This is an interesting work, dealing with health safety assessment in premises where occupants are either expecting some health improvement from their stay there or at least expect to live there in health secured environment.

The work provides a new vision of sanitary situation in targeted premises focused on chronical risks relating to inhalation exposure of indoor pollutant/toxic chemicals liable to exist in detectable quantities as identified in previous srudies carried out by the same research team.

There is however some margins from improvement considering  the broad readership of this journal by consideration of the following comments and suggestions.

  1. issue regarding what is said about formaldehyde.

As reported in the abstract and in  the result section, health risk regarding this chemical entails carcinogenicity: the reason why the sentence written in the introduction section saying "Hwang et al conducted a health risk assessment in hospitals, geriatric hospitals, elderly care facilities and postnatal care centers but only for non-carcenogenic risk due to exposure to formaldehyde": the way this sentence is written might lead some readers think that formaldehyde is no cancer risk, which of course is not true. I suggest to be more clear to avoid any confusion in the matter.

2. In relation with the previous point, I also suggest to mention the official health hazard rating for all the toxics of interest indicated in table 1 by detailing officially recognized health hazards (eg "acute tox, cat 2."..) in another table (either in section 2.1 or in supplementary information sheet or annex) , as derived from the so called (EU) Reach and/or CLP Regulations: this would clearly bring insightful information about what type(s) of chronical risks (cancer or not cancer related) in relation (or  not) to inhalation exposure (other types of exposure hence provided here just for information) have to be considered for the designated chemicals.

3. (editorial)
in section 2.1, second line: therr is an error message remaining as regard the expected cited references

4 discussion section

A number of studies have regarded inhalation exposure risks due to air indoor contamination in private dwellings. Indeed, it would be valuable to rank the level of risks as determined in your study as compared to what is known about the related risks pertaining to people exposure to same chemicals in private homes,  should enough pertinent information to exploit be available (may be notably in cited ref. 4 and some others that are not quoted so far) ?

Author Response

Responses to reviewer 1

This is an interesting work, dealing with health safety assessment in premises where occupants are either expecting some health improvement from their stay there or at least expect to live there in health secured environment.

The work provides a new vision of sanitary situation in targeted premises focused on chronical risks relating to inhalation exposure of indoor pollutant/toxic chemicals liable to exist in detectable quantities as identified in previous srudies carried out by the same research team.

There is however some margins from improvement considering the broad readership of this journal by consideration of the following comments and suggestions.

Thank you for your review and comments.

  1. issue regarding what is said about formaldehyde.

As reported in the abstract and in  the result section, health risk regarding this chemical entails carcinogenicity: the reason why the sentence written in the introduction section saying "Hwang et al conducted a health risk assessment in hospitals, geriatric hospitals, elderly care facilities and postnatal care centers but only for non-carcenogenic risk due to exposure to formaldehyde": the way this sentence is written might lead some readers think that formaldehyde is no cancer risk, which of course is not true. I suggest to be more clear to avoid any confusion in the matter.

The sentence has been rephrased. “Hwang et al. conducted a health risk assessment in hospitals, geriatric hospitals, elderly care facilities and postnatal care centers for non-carcinogenic risk due to exposure to formaldehyde. Although classified as carcinogenic, formaldehyde health risk assessment for carcinogenic risk has not been studied.”

  1. In relation with the previous point, I also suggest to mention the official health hazard rating for all the toxics of interest indicated in table 1 by detailing officially recognized health hazards (eg "acute tox, cat 2."..) in another table (either in section 2.1 or in supplementary information sheet or annex) , as derived from the so called (EU) Reach and/or CLP Regulations: this would clearly bring insightful information about what type(s) of chronical risks (cancer or not cancer related) in relation (or  not) to inhalation exposure (other types of exposure hence provided here just for information) have to be considered for the designated chemicals.

Health hazard were added in a supplementary table, as suggested.

  1. (editorial)
    in section 2.1, second line: therr is an error message remaining as regard the expected cited references

We corrected this mistake.

  1. discussion section

A number of studies have regarded inhalation exposure risks due to air indoor contamination in private dwellings. Indeed, it would be valuable to rank the level of risks as determined in your study as compared to what is known about the related risks pertaining to people exposure to same chemicals in private homes,  should enough pertinent information to exploit be available (may be notably in cited ref. 4 and some others that are not quoted so far) ?

We completed our discussion section with comparison to exposures and risk assessments performed in French dwellings. We added several references.

Regarding deterministic effects, we added the following sentences: In comparison with French dwellings, most of the VOCs and SVOCs were measured in higher concentrations at home [24,29]. Regarding SVOCs, the HQs were less than 1 [30].Regarding VOCs, their high concentrations in French dwellings were associated with an increasing prevalence of asthma and rhinitis in adults [31].

Regarding stochastic effects, we added the following sentences: In comparison with others European indoor environments, the ELCRs calculated in this study were lower [40]. In comparison with French dwellings, no ELCRs were greater than 1 x 10-5 for SVOCs [30]. In China dwellings, the highest ELCRs were found for formaldehyde (6.6 to 7.2 x 10−5) [41].

Reviewer 2 Report

The present work presents a quantitative health risk assessment in indoor places to workers due to 49 volatile/semi-volatile organic compounds. It concludes that only 3 substances presented HQ bigger than 0.1 (but smaller than 1). Another relevant conclusion is that only formaldehyde presented an excess lifetime cancer risk greater than 1 x 10-5 with the value of 3.8 x 10-5.

Despite I do agree that the sentence in lines 198 till 201 is a very likely explanation, I think it should be presented as a very likely hypothesis and not as a fact.

The main work “fragility” is assumed and well explained in the text (between lines 197 and 220). The importance of this issue is correctly addressed by authors but “toxics” editorial must consider this issue. Despite this, in my view, there are still significant scientific content/ data analysis.

Lines 83, 106, 169 and 189 have problems with reference sources “Error! Reference source not found”

Table S2 in "Supplementary Materials" has several chemical names with accentuated words (french language!?)

Author Response

Responses to reviewer 1

Thank you for your review and comments.

The present work presents a quantitative health risk assessment in indoor places to workers due to 49 volatile/semi-volatile organic compounds. It concludes that only 3 substances presented HQ bigger than 0.1 (but smaller than 1). Another relevant conclusion is that only formaldehyde presented an excess lifetime cancer risk greater than 1 x 10-5 with the value of 3.8 x 10-5.

Despite I do agree that the sentence in lines 198 till 201 is a very likely explanation, I think it should be presented as a very likely hypothesis and not as a fact.

The sentence has been rephrased. “Beyond potential differences in the practice and use of chemicals, ventilation systems in hospitals allows the renewal of the air more efficiently and most likely contribute to the reduction of the organic compounds concentrations. “

The main work “fragility” is assumed and well explained in the text (between lines 197 and 220). The importance of this issue is correctly addressed by authors but “toxics” editorial must consider this issue. Despite this, in my view, there are still significant scientific content/ data analysis.

Lines 83, 106, 169 and 189 have problems with reference sources “Error! Reference source not found”

We corrected these mistakes.

Table S2 in "Supplementary Materials" has several chemical names with accentuated words (french language!?)

We corrected these translation errors.

Author Response

Responses to reviewer 3

This manuscript calculated the health risk of VOCs and SVOCs exposure for occupants in healthcare and elderly care facilities in France. My recommendation is not to accept it before major revision.

Thank you for your review and comments.

  1. Line 83: (Error! Reference source not found.), please correct it.

We corrected this mistake.

  1. Line 169: (Error! Reference source not found.), please correct it.

We corrected this mistake.

  1. Line 189: (Error! Reference source not found.), please correct it.

We corrected this mistake.

  1. What is the “Figure 36” in Table 1?

We corrected this mistake.

  1. The authors should describe the related parameters about dose-response and exposure assessment in the supplemental material.

Toxicity and selected TRVs were added in a supplementary table, as suggested (Table S1).

  1. Although the authors said that the “Sampling and analyses were described in a previous paper [13]”, they still should simply describe the methods of sampling and analyses for these healthcare and elderly care facilities in the main text.

A paragraph regarding sampling and analyses was added in the materials and methods section (2.3. Exposure assessment).

  1. The authors selected 36 VOCs and 13 SVOCs according to activity and used chemical agents from reference 15 and found that the health risk for chronic inhalation of most compounds investigated did not appear to be of concern. Whether the activity and used chemical agents in hospitals emit these 36 VOCs and 13 VOCs and their potential sources (please provide related information in the main text)? If these VOCs or SVOCs are not presented in hospitals, it makes no sense for the authors to assess their health hazards.

In the introduction section, we added this paragraph, based on reference 17, to explain the choice of the molecules of interest.

“This methodology, developed by Berrubé et al., was based on a combination of classes of exposures and classes of hazard that produces a risk matrix. Criteria used to determine the exposure classes were quantity and frequency of the chemical's use, volatility and the type of collective protective equipment associated in each area. Criteria used to determine the hazard classes were classification as carcinogenic, mutagenic, or toxic to reproduction; the existence of a Toxicological Reference Values (TRV); occupational exposure limit val-ues; and risk phrases (official risk descriptions of the products)”

We agree with your comment: if an organic compound concentration was below the quantified threshold, its health risk assessment will not highlight a health hazard. However, we have chosen a standardized method using a worse-case scenario for all possible occupational exposures. We completed our materials and methods section (2.4. Risk characterization) as follow: “If an organic compound concentration measured was below the quantification threshold, this threshold was used to calculate the HQ (worse-case scenario).” 

Round 2

Reviewer 1 Report

I am satisfied with the efforts made to take account of my suggestions.

Reviewer 3 Report

No more question or comment.